# A cross-national study examining imaginary companions and face pareidolia in British and Chinese adults

Paige E. Davis[1]*, Charles Fernyhough[2], Liu Yang[1], Yijie Xi[1], Chenyu Xing[1], David Smailes[3]

1 Department of Psychology, University of Leeds, Leeds, United Kingdom, 2 Department of Psychology, Durham University, Durham, United Kingdom, 3 Department of Psychology, Northumbria University, Newcastle upon Tyne, United Kingdom

* p.e.davis@leeds.ac.uk

## Abstract

Although imaginary companions are created by children and sometimes adults around the world, the prevalence of this play behaviour varies. Cross-nationally, imaginary companions are reported more frequently in Western countries. These imaginary entities have been speculated to be similar to hallucination-like-experiences, based on evidence for elevated top-down auditory processing in children who report them. Face pareidolia tasks engage visual top-down processing, and performance on them does not tend to vary across cultures. This study asked if: 1) there would be cross-national differences in imaginary companion creation in childhood and adulthood between Chinese and British adults, 2) whether those creating imaginary companions would see more face pareidolia and 3) if there would be cross-national differences in face pareidolia. 291 participants (185 Chinese) completed a questionnaire on their imagination followed by a face pareidolia task consisting of 36 image trials (24 containing face pareidolia). Results showed that including all participants (Chinese and British) 11% of the adults currently had an imaginary companion. Chinese adults were significantly less likely than British adults to report a childhood, but not adulthood, imaginary companion. There were significantly more reports of face pareidolia from participants with a current imaginary companion, but not those who remembered a companion in childhood. The pareidolia hits did not differ between country, but false alarms were experienced significantly more by the Chinese participants. Taken together, the results provide more information around imaginary companion creation in China and the UK as well as the role top-down processing may play in imaginary companion interactions.

**Data availability statement:** The data associated with this manuscript can be found in the supporting files.

**Funding:** The author(s) received no specific funding for this work.

**Competing interests:** One of the authors, David Smailes is an editor at PLOS.

## Introduction

Imaginary companions (ICs) are a normative form of childhood play that can be found in different cultures throughout the world, although prevalence varies significantly between countries [1–3]. ICs were first described by Margret Svendsen (1934) as "An invisible character named and referred to in conversation with other persons or played with directly for a period of time, at least several months, having an air of reality for the child, but no apparent objective basis" ([4] p. 988). However currently personified objects (POs) such as dolls or toys are also placed in this category because they are argued to perform similar cognitive functions for children [5,6].

ICs are thought to be a normal childhood play experience and creators report that they are aware their ICs are not real [7,8]. Children as young as three years have been found to be able to distinguish fantasy from reality and youngsters with higher imaginative inclination have been found to be better at this cognitive capacity [9,10]. While invisible IC (iIC) creators understand that their playmates are not real, some report that they can see and hear them; however, this varies between individuals [8,11].

The current view is that ICs are a universal phenomenon occurring at a stage of development when children are playing and honing foundational cognitive skills. Researchers have investigated the prevalence of iICs and POs in different areas of the world [1]. Through a meta-analysis, Moriguchi and Todo (2022) found that the report of iICs (as opposed to POs) was more likely in the United States, Oceania and Europe than in Japan, with children in these locations reporting the creation of iICs up to 10 times more than Japanese children [1]. In a sample of Chinese 5–6 year olds, Lin et al. (2020) found a similar pattern to Japan, with 26.1% of the children reporting POs while 13.9% reported iICs [12].

Other research has focused on descriptions of ICs in Türkiye, where preschoolers were interviewed about their companions [13]. Yazici Arici et al. (2022) found six themes in the descriptions encompassing aspects of ICs like appearance, mood, communication and thinking [13]. Parental beliefs about ICs have been investigated in Mexico concluding that Mexican parents tend to be less approving and have more concerns about the play behaviour than those in America [8,14]. In India, Mills (2003) researched conceptualisations of ICs. She looked at how ICs created by American children compare to Indian children who report remembering their past lives [15]. No significant differences between these two groups were found in terms of when children first spoke of the phenomenon, how long it lasted, and how suggestable they were, amongst other variables.

Some studies have found that, in childhood, those with ICs tend to show heightened performance on tasks having to do with the understanding of mind (e.g., Davis et al., 2011; Taylor & Carlson, 1997), but only in some domains [16]. This elevated performance on tasks around theory of mind (the understanding that others can take different perspectives, as well as having different thoughts feelings or ideas) has been replicated in China, and has also been observed in adults that remember a childhood IC [12,17]. Furthermore, other between-group differences have been found to carry over into adulthood, such as propensity toward creativity [18].

These links between experiencing ICs and individual differences in theory of mind and creativity, as well as similarities in their phenomenology, have resulted in some researchers [(e.g., [19, 20])] suggesting that there are important parallels between IC and hallucinations. For example, when experiencing an IC or a hallucination, a person sometimes interacts with a social agent that others cannot see, hear, or interact with [(especially when the hallucination is auditory/verbal; 21)]. Thus, these two experiences seem phenomenologically similar. Meanwhile, both having an IC and having unusual sensory experiences (such as hallucinations) is associated with a range of overlapping cognitive-developmental processes [19] including greater self-reported creativity [22,21]. Thus, at least in part, a set of shared factors may play a causal role in the development of ICs and of hallucinations.

Consistent with these claims, evidence has been found that ICs can be an early indicator of developing non-pathological hallucinatory experiences [20,23]. Adults who previously had, or still had imaginary friends, were more prone to hallucinations than those without [20,24]. Thus, it seems plausible that some of the factors that play a role in the development of hallucinatory experiences will also play a role in the development of ICs, and there is some evidence consistent with this idea. For example, individual differences in the frequency of hallucinatory experiences (e.g., hearing a telephone ring when it has not) correlates with how often participants detect speech sounds in ambiguous auditory stimuli [25]. Meanwhile, children with ICs tend to detect more speech sounds in ambiguous auditory stimuli than do children without ICs [23,26]. If we assume (as others – e.g., [27] – have) that a person's tendency to detect speech sounds in ambiguous auditory stimuli reflect individual differences in auditory top-down processing (processing using prior knowledge or expectations to interpret ambiguous stimuli), then these findings suggest that elevated auditory top-down processing plays a role in the development of both ICs and hallucinatory experiences.

A number of studies have shown that ICs have a visual as well as an auditory component [8,28]. Thus, it would be valuable to examine whether visual top-down processing is associated with ICs. In hallucination research, visual top-down processing has been assessed using a face-pareidolia task [29]. In this task, participants are presented, for a short period of time, a series of images, most of which contain a face pareidolia (e.g., a picture of a cup of coffee where the foam at the top of the coffee 'looks' like a face). The remaining images are of complex scenes/objects, but do not contain face-like patterns. Participants are asked to judge whether each image contains a face pareidolia or not, and a stronger ability to detect face pareidolia is thought to reflect elevated top-down processing [30].

In previous work, individual differences in the ability to detect face pareidolia have been found to correlate with the frequency of visual hallucinatory experiences (e.g., seeing a person/animal in one's peripheral vision, even though no person/animal is there) in non-clinical participants [29]. It seems likely, therefore, that the ability to detect face pareidolia will also be associated with IC status; examining if this is the case was one aim of this study.

In addition, we aimed to examine whether any association between IC status and individual differences in the ability to detect face pareidolia was stable across cultures. Face processing shows pronounced cultural variation between East Asian and "Western" individuals, where Asians are more focused on holistic global information and "Westerners" are analytically focused on more salient features [31,32]. However, because face pareidolia tasks ask participants to view pictures and make assessments of objects resembling faces, the images/stimuli being processed are less complex than real faces, and so cross-cultural differences may not be found when studying face pareidolia. Consistent with this claim, Romagnano and colleagues reported an absence of cross-cultural differences in face pareidolia detection in a recent study [33]. However, Romagnano's research involved comparisons between participant groups from two European cultures which are relatively similar in geographical terms. There have not been any studies, to our knowledge, on individual differences in pareidolia in Chinese and British Adults, and our study aimed to investigate if any cross-national differences exist.

The aims of this study were to examine relationships between cross-national status, imaginary companion creation and face pareidolia in adulthood. Children were not included because of the investigatory nature of the study and the method of delivery being an internet platform. The research questions were:

1) Would there be significant cross-national differences between adult report of IC creation in childhood and adulthood between UK and Chinese participants?

2) Would there be significant differences between Chinese and British adults scores on a face pareidolia task?

3) Would there be significant differences between pareidolia scores based on adults' IC status?

Based on the previous research around cross-national prevalence of children's IC creation, we hypothesised that there would be a comparable number of ICs reported in England and China; however, we expected that Chinese participants would report creating more POs than those in the UK [1,12]. Because pareidolia is seen in non-face objects, and is sometimes used to rule out own-culture gender biases, we expected that there would be no significant cross-national differences in pareidolia scores [33]. We also expected that those with ICs would score higher on face detection in the pareidolia task as children and adults with ICs have 1) been found to be more susceptible to detecting patterns in verbal stimuli eliciting a similar hallucination like experience in a different modality [23,34]. Furthermore, 2) adults remembering ICs have been found to score higher on measures of hallucination, and report of pareidolia and hallucination are related [20,29].

## Materials and methods

### Participants

There were 291 participants recruited. Of the participants 92 (31.6%) identified as male, 192 (66%) identified as female and 7 (2.4%) responded other. When broken down by country, there were 185 Chinese participants. Only 180 (128 females) reported gender. There were 106 British participants. Only 104 (64 females) reported gender. Age and gender data for Chinese, British and total participants can be found in Table 1. Participants were not screened for any psychological, developmental or neurological conditions.

### Procedure

All project materials were prepared in English and Mandarin Chinese. The materials were originally created in English, so the translations were made by three researchers who, through discussion and drafting attempted to find the most accurate language equivalencies. All tasks remained the same with the exception of the associated language (Mandarin Chinese or English).

**Table 1. Participant demographics.**

| Age | Gender | China *N* (percentage) | England *N* (percentage) | Total *N* (percentage) |
|---|---|---|---|---|
| 18-30 Years | Male | 37 | 18 | 55 |
| | Female | 107 | 34 | 141 |
| | Total | 149 (80%) | 52 (49.1%) | 201 (69.1%) |
| 31-50 Years | Male | 12 | 19 | 31 |
| | Female | 20 | 26 | 46 |
| | Total | 32 (17.3%) | 47 (44.3%) | 79 (27.1%) |
| 50 + Years | Male | 3 | 3 | 6 |
| | Female | 1 | 4 | 5 |
| | Total | 4 (2.2%) | 7 (6.6%) | 11 (3.8%) |
| | Totals combined | 185 (100%) | 106 (100%) | 291 (100%) |

*Note* Age data were collected through ticking one of the 3 age categories. Totals include all participants ticking "prefer not to say" for the gender question.

Participants were recruited through social media platforms (Facebook in the UK and WeChat and QQ in China), through posters, word of mouth, the universities' participant pool in the UK, and the Prolific platform. Those using Prolific received nominal payment for their participation. Prolific has been found to produce high quality data sets with diverse participants [35]. Those recruited through the participant pool received one credit toward their module. Those not recruited through Prolific or the participant pool received a chance to be put into a prize draw for vouchers.

Participants we.re directed toward the Gorilla experiment builder platform, a platform which enables researchers to gain access to diverse samples of participants that might not normally be able to come to a laboratory setting [36]. The Gorilla link opens on the study's information sheet which explained the research. This was followed by an informed consent form. Once consent was given, participants were directed to the questionnaire. After the questionnaire a randomizer was used to direct the participant either to the face pareidolia task or another task focused on animacy which is not reported in this study. Once participants finished both face pareidolia and animacy tasks they were thanked for their time. All participants completed both tasks. Task order was randomized. The whole experiment took between 20–25 minutes.

The study was approved by the University of Leeds ethics committee on 07/03/2024, ethics code PSCETHS-1006. The project recruitment began on the 04/05/2024 and ended on the 20/06/2024. There were no affiliated Chinese institutions that ethically approved this study.

## Measures

**Imagination questionnaire [(Based on 20,37)].** Participants completed a questionnaire either in Chinese or English depending on the country of distribution. The first section asked their gender and age. Following these demographics, there was a definition of an IC according to the study's parameters which read: *an imaginary friend can be completely invisible or a doll or a toy which a child or adult has given a personality to and played with for at least 3 months*. After the definition was read there were 12 questions about ICs. The first 3 indicated whether the participant had a past or current IC. Those with an IC were then asked whether they hear, see or have any other sensory experiences with the companion. These questions were scored on a 5-point frequency scale from 0 = never to 4 = all the time. They were then asked whether the IC can act of its own accord and if it was a PO or completely invisible, what they liked and did not like about them and if the companion could boss them around. The final two questions asked on a scale of 1–10 how creative participants thought they were and how creative they were able to be in their job as creativity has been found to link to childhood IC status [18,38]. For examples of ICs see Table 2.

**Pareidolia task [29].** The face pareidolia task designed by Smailes and colleagues was used to measure top-down processing, and was presented to the participants through the Gorilla platform. Details of how the task was developed are provided in the Supplementary Materials. The task consists of 36 image trials, 24 of which contain images with face pareidolia and the remaining 12 have no pareidolia. Images were presented to participants in a random order. At the beginning of the task participants were informed that they would see some pictures that appear like they contain faces and others will not. The start of a trial began when the participant pressed a button on the screen saying "next". A fixation point would appear followed by a 'pareidolia present' or 'pareidolia absent' image for 0.75 seconds. This screen was followed by another with only the words "face" and "no face". Participants would be able to respond and move on to the next image after clicking their choice. As in previous work [29], we were primarily interested in the number of hits made on this task (where participants correctly responded that a face-pareidolia was present in the image). Participants' scores on this measure could range from zero to 24, However, we also recorded number of false alarms made (where participants incorrectly reported that a face-pareidolia was present in the image). Participants' scores on this measure could range from zero to 12. For examples see Fig 1. Following Parsons et al. (2019), we estimated the internal reliability of this measure [39]. We did so using RELEX [40], an Excel-based tool that calculates a permutation-based split-half reliability for measures obtained by tasks. We found that the primary measure we obtained from the task (number of hits made) had acceptable levels of reliability (median rSB = .89, 95% HDI 0.85–0.92). However, we found that the second measure we

**Table 2. Examples of iICs and POs in China and the UK.**

| Country | IC Type | Description |
|---|---|---|
| China | iIC | A white mist-like creature that often floats in the air and can change its shape. I can always play simulation games with her when I am alone, she is very patient and likes to share. She can be overly enthusiastic sometimes, interrupting me when I'm busy with other things and making me lose focus. |
| | iIC | A sophisticated woman with an elegant and mature style, who likes cool Japanese school uniforms. Completely feminine, she smokes and drinks, is bisexual, knowledgeable, composed, and decisive when dealing with situations. She can provide me with guidance, similar to the function of the superego in psychology. She's so good that sometimes she gets out of my control. |
| | PO | The lion doll from when I was a kid. It has always quietly accompanied me. It was with me when I encountered unhappy things during my childhood. |
| | PO | A rabbit doll. It always listens to me and gives me room to talk. My dislikes are that it can't respond to me. |
| United Kingdom | iIC | He looked like a caricature of an old sea captain, short and slightly chubby, with a white beard, a cap, an old suit, and a pipe in his mouth. |
| | iIC | more of a voice in my head to be around. gave me company as a kid and was someone to interact and play with. i liked the fact that it gave me company when feeling lonely. |
| | PO | She was an old doll made of ceramics, with a very untidy black wig. I could tell her things, and share feelings. My doll was an outlet. She had a gap between the back of her head and her body. If I wasn't careful, she would "bite" my fingers. |
| | PO | a brown teddy that had button eyes and is around hand sized. he offered comfort whenever I moved places or went abroad etc because he reminded me of home. he was sometimes mischievous and would go places where he wasn't supposed to. |

*Note iIC = invisible IC, PO = personified object. Descriptions are taken straight from the questionnaire with no changes except for the statement "My dislikes are" for the 2nd Chinese PO. The Chinese participants' reports were translated and agreed upon by the three Chinese researchers.

obtained from the task (number of false alarms made) had lower levels of reliability (median $rSB = .66$, 95% HDI 0.58–0.74). A version of the task is openly available at https://osf.io/g3jac via doi.org/10.17605/osf.io/t7vdc.

## Results

### Descriptive statistics

**IC prevalence across countries.** First, IC status was investigated. Out of the 291 participants 142 (48.8%) reported having a childhood or adulthood IC. Of the participants reporting ICs at some point in their lives 127 (44.6%) created one in childhood and 31(11%) currently had one.

Dividing the group between countries, we examined IC status in reference to country finding that through life significantly more ICs are being made by those in the UK $\chi^2$ (1, $N=291$) = 10.47, $p=0.001$, $\varphi=0.19$. This also holds true for past ICs where UK participants report significantly more $\chi^2$ (1, $N=291$) = 5.71, $p=0.017$, $\varphi=0.14$. But no differences were found between countries in having current ICs $\chi^2$ (1, $N=291$) = 3.01, $p=0.083$.

When IC status was broken down into PO and invisible IC report through life 49 (35.8%) were invisible and 88 (64.2%) were personified and 5 were removed from the analysis as they were not specified. Significantly more POs were reported in Chinese participants than in UK participants $\chi^2$ (1, $N=137$) = 16.82, $p<.001$, $\varphi=0.35$, and this was also true for childhood POs in China $\chi^2$ (1, $N=132$) = 16.15, $p<.001$, $\varphi=0.35$, but there were no differences in adulthood between IC and PO creation (Fisher's Exact test = .226). For more details of IC status according to country see Table 3.

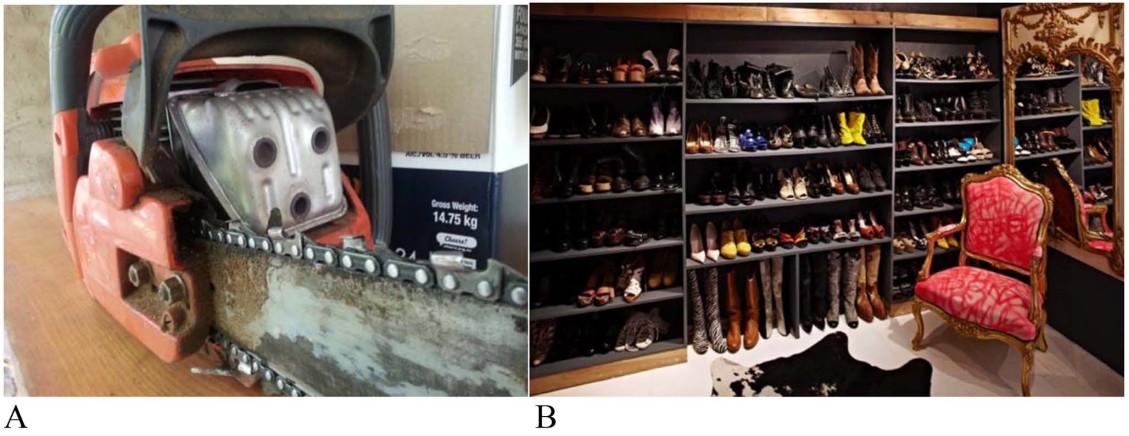

A    B

face    no face

Next

C

**Fig 1. Example of an image trial with face pareidolia A and without face pareidolia B and response screen C.** *Note.* After each image trial participants are presented with a response screen. If a participant were to report a "face" for image trial A this would indicate a hit. If they were to report "no face" it would count as a miss. For image trial B, the report of a "face" would be considered a false alarm and "no face" would be a correct rejection.

**Table 3. Prevalence of ICs in China and the UK.**

| Country | IC Status | Childhood | Current |
|---|---|---|---|
| China | No IC | 112 (60.5%) | 158 (85.4%) |
| | iIC | 15 (8.1%) | 6 (3.2%) |
| | PO | 58 (31.4%) | 21 (11.4%) |
| | Total IC | 73 (39.5%) | 27 (14.6%) |
| United Kingdom | No IC | 47 (44.3%) | 97 (91.5%) |
| | iIC | 32 (30.2%) | 4 (3.8%) |
| | PO | 27 (25.5%) | 5 (4.7%) |
| | Total IC | 59 (55.7%) | 9 (8.5%) |

*Note* IC = imaginary compaion, iIC = invisible imaginary compaion, PO = personified object.

**Gender and ICs.** Gender was considered as a potential variable which could impact IC status. Overall, there were no significant gender differences between IC creators over childhood and presently $\chi^2$ (1, $N$ = 284) = 0.36, $p$ = 0.551. There was also no difference when looking at China alone $\chi^2$ (1, $N$ = 180) = 0.79, $p$ = 0.374 and the UK alone $\chi^2$ (1, $N$ = 104) = 0.17, $p$ = 0.677.

## Main analyses

**Pareidolia and ICs.** Participants' scores on the pareidolia task (for both number of hits and number of false alarms) were not normally distributed. Descriptive statistics are presented in Table 4.

We tested whether the differences between the IC Groups were significant using a pair of Mann-Whitney U tests. The difference in the number of Pareidolia False Alarms between the Group without a Childhood IC (median = 0, IQR = 1) and the Group with a Childhood IC (median = 0, IQR = 1) was not significant, $U$ = 9456, $z$ = 0.96, $p$ = .336, $r$ = .06. Similarly, the difference in the number of Pareidolia Hits between the Group without an IC (median = 16, IQR = 7) and the Group with an IC (median = 17, IQR = 7) was not significant, $U$ = 9326.50, $z$ = 1.02, $p$ = .306, $r$ = .06.

The difference in the number of Pareidolia False Alarms between the Group without a Current IC (median = 0, IQR = 1) and the Group with a Current IC (median = 0, IQR = 2) was not significant, $U$ = 3421.50, $z$ = 1.30, $p$ = .192, $r$ = .08. In contrast, the difference in the number of Pareidolia Hits between the Group without a Current IC (median = 16, IQR = 7) and the Group with a Current IC (median = 18, IQR = 8) was significant, $U$ = 2773.50, $z$ = 2.64, $p$ = .008, $r$ = .16.

Given the nature of the pareidolia task, we were also able to calculate signal detection parameters, using participants' responses. We calculated $d'$ (or sensitivity – that is, how well a participant discriminated between trials where a face pareidolia was present and trials when there was no face pareidolia) and $C$ (or criterion – that is, to what extent was a participant biased towards that a face pareidolia was present, regardless of whether one was present or absent). Analyses using these two outcomes is presented in the Supplementary Materials. However, those findings are consistent with the findings above. Group differences were not significant when we compared participants with versus without a childhood IC. Similarly, the group difference in $d'$ was not significant when we compared participants with versus without a current IC. However, there was a significant group difference in $C$ when we compared participants with versus without a current IC. Consistent with the greater number of 'hits' made by participants with a current IC, they also had a lower median $C$-score (i.e., they had a more liberal bias, towards responding that a face was present in the images).

Our assessment of the nature of participants' ICs allowed us to examine associations between performance on the pareidolia task and individual differences in, for example, the 'sensory' experiences of an IC (e.g., whether the participant reported seeing or hearing their IC). This analysis was exploratory, and these associations are reported in Supplementary Materials.

**Pareidolia and culture.** We tested whether there were significant cross-cultural differences in task performance using Mann-Whitney U tests. The difference in the number of Pareidolia Hits between the Chinese participants (median = 16, IQR = 7) and the British participants (median = 16, IQR = 6) was not significant, $U$ = 9642, $z$ = 0.24, $p$ = .813, $r$ = .01. However, the difference in the number of Pareidolia False Alarms between the Chinese participants (median = 0, IQR = 1) and the British participants (median = 0, IQR = 0) was significant, $U$ = 7456, $z$ = 3.92, $p$ < .001, $r$ = .23, with Chinese participants making significantly more false alarms on the task.

For more information on intercorrelations, and scores on imaginativeness please see supplementary materials.

**Table 4. Performance on the Pareidolia Task across IC groups.**

|  | Full Sample | Group without Childhood IC | Group with Childhood IC | Group without Current IC | Group with Current IC |
|---|---|---|---|---|---|
| Mean Pareidolia False Alarms (SD) | 0.74 (1.34) | 0.77 (1.35) | 0.70 (1.34) | 0.70 (1.33) | 1.00 (1.46) |
| Mean Pareidolia Hits (SD) | 14.81 (5.68) | 14.47 (5.75) | 15.09 (5.69) | 14.53 (5.70) | 16.81 (5.76) |

## Discussion

We set out to determine whether there were cross-national differences in adult reports of IC creation and type (IC or PO) between participants from China and the UK. We also wanted to know whether there were differences in the report of face pareidolia between the two countries as well as whether IC status impacted pareidolia scores.

Results showed that overall, almost half of the participants reported creating childhood ICs and 11% reported current ICs. Participants in the UK created significantly more ICs in childhood and those ICs were more likely to be iICs than their Chinese counterparts. These results did not hold true for participants' current reports of IC prevalence and IC type. The pareidolia results showed that remembering a past IC was not related to pareidolia detection or false alarms, however those with current ICs in both China and the UK showed significantly more pareidolia hits. Chinese participants had more false alarms than British participants, but no differences were found between countries when considering pareidolia hits.

IC studies looking at children range in prevalence rates depending on the inclusion and exclusion criteria as well as the populations, assessment method and ages of participants [1,7,8]. This is the first cross-national study of Chinese and British adults, so we were unsure of what prevalence differences between countries might arise, but suspected that there would be none based on Moriguchi and Todo's (2022) meta-analysis [1]. Moriguchi and Todo investigated IC prevalence between cultures finding no significant differences between countries, however they focused on Europe, Oceania, the United States and Japan and did not include Chinese studies.

There have been studies into children with ICs in China and in the UK, but none that make comparisons between the two [12,41–44]. Lin and colleagues [42] found that 36.8% of the parents participating in their study reported that their Chinese child had an imaginary companion, while Davis and colleagues (2013) had 46% of the children in their study report an IC – however, there was a discrepancy between parent and child reports where parents reported less than their children. In our research, British adults remembered significantly more ICs in childhood than Chinese adults. One possible explanation for this difference could be that British culture has been traditionally considered to be individualistic, valuing independence, novelty, autonomy and freedom, all values which may lead to creation of an imagined social relationship [45]. Collectivist cultures like China have traditionally been more likely to value harmony, conformity and consensus, which may result in less individual ICs [45]. However, with more movement around the globe, these differences may disappear/ lessen. Another possibility is that the questionnaire focused on remembering an IC rather than actually having one. Retrospective report of an IC has been found to be inaccurate in adolescents and misremembering may be a reason to interpret the results around childhood ICs with caution [46].

Adult report of a current IC was higher than we had anticipated at 11% and there were no between-country differences in reporting. This statistic is higher than that of Fernyhough and colleagues' (2019) finding that 7.5% of the adults surveyed reported current ICs. Their study was carried out in the UK and advertised in a newspaper and at a book festival, so may have gleaned different audiences than our study which recruited mainly through social media in both countries. Recruitment taglines stated participants needed. There was no mention of ICs in the recruitment, however the body of the advertisement said that the researchers were investigating imagination and face perception which may have steered those more interested in imagination. Although it has been found to be a robust platform, using prolific for recruitment in the UK may have gleaned a different group of respondents than pure social media recruitment as Prolific participants are paid a small amount for their participation, whereas in China there was simply a prize draw for a possible gift card.

The results around Chinese participants reporting significantly more childhood POs than British participants was consistent with previous research on the form of Eastern and Western children's ICs [1,42]. Although in childhood Chinese participants differed in their report of IC type, this discrepancy disappeared for those with current ICs. Gleason and colleagues [47] found that American children with POs were more inclined to care for them while those with iICs were more likely to have egalitarian relationships. It would be interesting to know whether this difference in relationship quality carries over to the UK and China, and whether the phenomena itself could be a cultural variation.

 

When considering the pareidolia scores, there were cross-national differences in false alarms where Chinese participants made significantly more false alarms, but there were no differences in pareidolia hits. It is well documented that there are cultural differences in face processing [(e.g., 46)]. For example, in Europe, French-speaking Swiss participants and German participants had different patterns of face tuning [48]. More diverse culture shifts comparing Canadian and Chinese participants show that even the way visual information about a face is perceived can be impacted by culture [49,50]. It may follow that, with these cultural differences in face processing, there may also translate to differences in pareidolia perception.

We think that this is an unlikely explanation, as one would expect that any cultural differences in face-processing would lead to cross-national differences in both face pareidolia hits and face pareidolia false alarms in our dataset. Instead, it is possible that this cross-national difference in the number of false alarms made reflects differences in processes unrelated to face-processing or pareidolia. That is, our conceptualization of the face pareidolia task is that making a large number of hits is associated with elevated top-down processing, but that making a large number of false alarms is not (the close-to-zero and moderate correlations between number of hits and number of false alarms made on the task in [29], and in the present study, respectively, are consistent with this conceptualization). Thus, the greater number of false alarms made by Chinese participants may not reflect differences in face processing or in processes related to face-pareidolia, but may instead reflect cross-national differences in general response biases [51] or in metacognitive confidence [52] At present it is unclear how general response biases (such as acquiescence) and/or metacognitive confidence relate to performance on a face pareidolia task (e.g., if they are associated with hits and false alarms in different ways), and so it is not clear how plausible this claim is. Finally, it is possible that this cross-national difference in number of false alarms made may simply be statistical 'noise' (i.e., it may be a Type I error), rather than a reflection of cross-cultural differences in cognition. While, the $p$-value associated with this effect was very small and some may infer, therefore, that it is very unlikely that this is a Type I, a single small $p$-value is as likely to occur as a larger $p$-value (e.g.,.04 or.99) when the null hypothesis is true [(as p-values are uniformly distributed when the null hypothesis is true; 53)] Future research, would be well placed to examine whether general response biases and/or metacognitive confidence ratings could explain group differences. It would also be beneficial to replicate the cross-national difference found in this study.

Some researchers suggest that those experiencing ICs also have higher likelihood of a profile rooted in transliminality [21]. Transliminality refers to thin mental boundaries between conscious and unconscious where an individual may involuntarily be more susceptible to inwardly generated thoughts [54]. This has been found to relate to sensing presence and hallucination derived from a range of scales [55].

The improved pattern detection (i.e., greater number of hits on the pareidolia task) of those with current ICs is consistent with the idea that elevated top-down processing may play a role in the development of ICs. However, the finding that performance on the pareidolia task was associated only with the presence of a current, and not a childhood IC suggests that this could be a dynamic, state-like association rather than something that is highly stable as a person ages. More research would need to be done to support this theory, as we do not have longitudinal data and only retrospective accounts. The present findings concerning elevated top-down processing in the visual modality fits into the current literature around people with ICs hearing more words in the auditory task [23,34]. Because ICs can be iICs or POs and both types can have any combination of auditory and visual components for their creators, there are questions around whether there are related differences in face or auditory verbal detection depending on whether one sees or hears their companion. It might be interesting for future research to investigate modality of IC experience and determine whether this relates to top-down processing in only visual tasks, but also auditory. It could also be interesting to parse apart the iICs from the POs, however these analyses would need many more adults. Parsing apart iICs from POs is especially important with the number of POs in the study sample. Because POs can be seen and may not relate as much to top-down processing, there could be another reason for the results.

Another consideration around iICs and POs is that the experience children are having with them might later result in differences in pareidolia. We theorise that children are using the same type of imagination processing with iICs as with

POs. As Taylor and Aguiar argue, children might very well be utilizing their own imagination to embellish the appearance of their PO so that it may look quite different in their mind's eye than in reality [8]. Asking participants for details around what their PO looks like to them might be a question to ask adult participants in future research. If it were true that iICs and POs were different imagination processing experiences, the results of many IC studies including this one would need to be reconsidered.

Looking where in the world the phenomena of IC creation is occurring most often in children would also generate important information around whether this top-down processing is developed through current ICs. Age is one variable to consider in researching pareidolia-proneness, as this could change as people age, in the same way that proneness to hear speech in ambiguous noise reduces as people age [27]. Similarly, and perhaps related to this, the tendency to hallucinate typically reduces from adolescence into adulthood [56]. Looking at younger children who are also more likely to have hallucination-like experiences might help to unpick whether modality and age impacts pareidolia.

There are limitations to this study that need to be considered. The first is that with only 11% of the participants reporting ICs at present, we may have failed to find associations/group differences due to a lack of power (e.g., this seems possible with reference to the absence of a group difference in the prevalence of current ICs), and some of our effect size estimates will be imprecise. Recruiting participants for a larger data set would help to improve this in future. Furthermore, adults retrospective reports of ICs could also pose a problem as memory of past ICs fades even as early as adolescence [46]. Having children experiencing ICs while participating would help to determine if there are similar results. The age of the participants was taken in ranges rather than having participants give discrete numbers for their age. Although there were no differences in age groups (see appendices), it would be useful to be able to look at more fine-grained differences in age, especially because 80% of the Chinese sample was between 18–30. Another consideration is that participants were not screened for any psychiatric, developmental or neurological conditions and given the significant impact on social cognition and perceptual processes, having this information could improve our ability to evaluate the results. Finally, translations from English to Mandarin were not reviewed by a professional translator, however the three researchers that translated the instructions were natives to China and thus believe that between the three of them there was an accurate translation.

On the whole, this study provides more evidence consistent with the notion that ICs are found throughout the world but may differ in form and rate of production in childhood. It also provides more evidence that with the cognitive processes involved in the development of hallucinations may also be involved in the development of ICs. Further research that has greater statistical power, and that examines whether there are associations between elevated top-down processing in specific sensory modalities and IC experiences in that same modality, would be valuable.

## Supporting information

**S1 File. Supplementary materials, analyses and tables for A Cross-National Study Examining Imaginary Companions and Face Pareidolia in British and Chinese Adults.**
(DOCX)

**S2 File. Plos One inclusivity in global research questionnaire.**
(DOCX)

**S3 File. SPSS data file with C and D prime scores.**
(SAV)

## Author contributions

**Conceptualization:** Paige Davis.

**Data curation:** Liu Yang, Yijie Xi, Chenyu Xing.

**Formal analysis:** Paige Davis, Charles Fernyhough, David Smailes.

**Investigation:** Paige Davis, Liu Yang, Yijie Xi, Chenyu Xing.

**Methodology:** Paige Davis, David Smailes.

**Project administration:** Paige Davis.

**Supervision:** Paige Davis.

**Writing – original draft:** Paige Davis, Charles Fernyhough, David Smailes.

**Writing – review & editing:** Paige Davis, Charles Fernyhough, David Smailes.

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
