## [Decision Letter · Decision Letter 0]

6 Aug 2025

A Cross-National Study Examining Imaginary Companions and Face Pareidolia in British and Chinese Adults

PLOS ONE

Dear Dr. Davis,

Thank you for submitting your manuscript to PLOS ONE. After careful consideration, we feel that it has merit but does not fully meet PLOS ONE’s publication criteria as it currently stands. Therefore, we invite you to submit a revised version of the manuscript that addresses the points raised during the review process.

I agree with the reviewers that it is crucial to reflect on the selection of the face pareidolia task in particular, and how this measure is best interpreted, especially in relation to discussion around visual hallucinations. Addressing the points raised about this are likely to result in significant changes to the manuscript. Please ensure to carefully address all comments raised by reviewers.

We look forward to receiving your revised manuscript.

Kind regards,

Clare Eddy

Academic Editor

PLOS ONE

Journal Requirements:

[One of the authors, David Smailes is an editor at PLOS.].

4. Please include a complete copy of PLOS’ questionnaire on inclusivity in global research in your revised manuscript. Our policy for research in this area aims to improve transparency in the reporting of research performed outside of researchers’ own country or community. The policy applies to researchers who have travelled to a different country to conduct research, research with Indigenous populations or their lands, and research on cultural artefacts. The questionnaire can also be requested at the journal’s discretion for any other submissions, even if these conditions are not met.  Please find more information on the policy and a link to download a blank copy of the questionnaire here: https://journals.plos.org/plosone/s/best-practices-in-research-reporting. Please upload a completed version of your questionnaire as Supporting Information when you resubmit your manuscript.

6. We are unable to open your Supporting Information file [Pareidolia Cross-national PLOS.sav]. Please kindly revise as necessary and re-upload.

Reviewers' comments:

Reviewer's Responses to Questions

**Comments to the Author**

1. Is the manuscript technically sound, and do the data support the conclusions?

Reviewer #1: Yes

Reviewer #2: Yes

Reviewer #3: Partly

2. Has the statistical analysis been performed appropriately and rigorously?

Reviewer #1: Yes

Reviewer #2: Yes

Reviewer #3: Yes

3. Have the authors made all data underlying the findings in their manuscript fully available?

Reviewer #1: Yes

Reviewer #2: Yes

Reviewer #3: Yes

4. Is the manuscript presented in an intelligible fashion and written in standard English?

Reviewer #1: Yes

Reviewer #2: Yes

Reviewer #3: Yes

Reviewer #1: I found this to be an interesting study exploring the influence of culture and age on the formation of imaginary companions (ICs) and their relationship with face pareidolia perception. The statistical analyses appear to be appropriately conducted, and the authors clearly acknowledge the study’s limitations in the discussion, such as the small sample size in the group with current IC experiences. However, there are a few additional points that could be addressed or elaborated further:

Abstract:

The conclusion of the abstract comes across as too broad and somewhat disconnected from the actual study scope. For example, the sentence: “Taken together, the results provide more information around the role of top-down processing in both imaginary companion creation and hallucination. This could be a way to learn about the developmental trajectory of hallucination from non-pathological to pathological.” seems overly speculative. Since the study is mainly of exploratory nature and focuses on ICs and face pareidolia, it does not directly investigate hallucinations, especially not pathological ones. For instance, Romagnano et al. (2022) examined face pareidolia in individuals with schizophrenia and found no significant differences in pareidolia perception between patients with or without hallucinations.

Participants:

It would be helpful to clarify whether participants were screened in the questionnaire for psychiatric, developmental, or neurological conditions, given the significant impact such conditions can have on social cognition and related perceptual processes.

Visualization:

The paper would benefit from improved visualization. Including examples of the stimuli used and/or graphical representations of key results would help readers better understand and interpret the findings.

Discussion:

The authors mention that they were surprised to find cultural differences between the UK and China in face pareidolia, referencing a study that found no significant differences between two European subcultures (reference 33). However, given the well-documented differences in face processing between Western and East Asian populations, differences in pareidolia perception might indeed be expected. This could be discussed in more depth, potentially referencing additional literature.

Additionally, Zhou and Meng (2021) analyzed face pareidolia using signal detection theory (SDT). Incorporating similar analyses, such as sensitivity (d') and decision criterion (c), could provide more insight into the nature of the cultural differences observed and should be considered either in this paper or in future work.

Reviewer #2: Note: No generative AI tools were used in the reading and reviewing of this manuscript. Thus, the contents of this review are my own subjective take on this manuscript and were authored by me, the reviewer. Therefore, if there are typos/errors in this review, I apologize. I am only human, after all.

Strengths: This study adds to the growing body of evidence that imaginary companions (ICs) exist across cultures and is a phenomenon that extends beyond the early childhood years. This study further replicates existing research indicating that propensity to create general types of ICs can vary across cultures (i.e., invisible companions or personified objects). Finally, this study also adds to the literature on phenomenon related to the creation of ICs, namely the ability to see/detect faces in visual imagery. It therefore makes a valuable contribution to the extant literature.

Opportunities: There are a number of opportunities to improve this manuscript. I address the most important suggestions first, followed by other comments that I hope are helpful.

Overall Challenges

I am struggling to understand the use of face pareidolia as a proxy for hallucinations, particularly given the task that was used to measure face pareidolia in this study. Hallucinations, to the best of my understanding, are hearing, seeing or experiencing things for which there is no basis of reality, or for which there is no objective evidence of existence. With hallucinations, what is “real” or “true” appears to be a key consideration in measuring what is and is not a hallucination (at least in western cultures). However, seeing faces in visual stimuli such as abstract images, clouds, objects, etc. is a subjective experience in which a basis of reality may not matter as much, and to the best of my knowledge, more closely associated with individual differences in anthropomorphism in the literature. With abstract imagery, the presence of a face in visual stimuli could simply be “in the eyes of the beholder.” In other words, seeing a face in abstract visual imagery is neither “true” nor “false”; it just is.

And that brings me to my concern about the task designed to measure face pareidolia, where “correct hits” are described as suggestive of a greater propensity for hallucinations, and “false positives” are discussed simply as errors. If the task was designed as a measure or proxy for the experience of hallucinations, then the “false positives” are of greater interest than the “correct hits.” And in this study, there were no differences in false positives based on IC status (past or current, absent or present), which would change the authors’ interpretation of the findings. However, if the task is not intended to serve a proxy for hallucinations but rather a direct measure of face pareidolia alone, then it is unclear why the measure needed to have “correct” or “incorrect” responses. In other words, even with the same task, why not simply measure for each participant the number of images in which a face was detected?

Suggestions: In the literature review, make a stronger case as to how seeing faces in visual stimuli has links to hallucinations and to ICs, from a theoretical perceptive and in terms of measurement. Reconsider the description of the measure of face pareidolia (i.e., if it is indeed being used as a proxy for measuring hallucinations or if is not meant to imply anything beyond face pareidolia). If it is being used as a proxy for measuring hallucinations, please explain why “false positives” were not the primary variable of interest.

Literature Review

Again, in the literature review, I am struggling to understand how top-down processing, ICs, hallucinations, and face pareidolia are all theoretically connected. Consider adding a schematic image to clarify for the reader.

The literature review also assumes prior knowledge of other ideas/constructs, such as theory of mind and top-down processing. Provide brief definitions so that readers who are not as familiar with this literature can understand the points that are being made.

In lines 98 – 102, it would be helpful to understand the interplay between cross-cultural differences in face pareidolia and ICs. Perhaps speculate as to what this interaction might be or look like in this section. In addition, more information about how face pareidolia tasks have been used to assess cultural and gender biases would be helpful.

Also, in the literature review, justify why children were not included in this study or the focus of the study, as retrospective reports of ICs are problematic due to memory/recall, and research question 1 (as it is written) suggest that adult and child reports of ICs will be compared.

Finally, throughout the manuscript, please take care in discussing findings using developmental language, as this was a study of adults and did not make direct comparisons with children, or follow participant development over a period of time (see lines 322 – 323 as an example).

Method and Materials

In lines 146-148, please clarify if participants completed both the face pareidolia task and the animacy task in randomized order, or whether they were randomly assigned to complete only one of these tasks. If the latter in the case, then clarify the number of participants in the whole sample and those that completed the face pareidolia task that were used in the analyses.

In lines 150 – 152, clarify if any ethics committees in China were involved in approving this study.

In line 178, please report timing in seconds rather than milliseconds for better reader comprehension.

In lines 191 – 192, thank you for providing a link to the task. However, one figure showing a “correct hit” and a “false positive” would be helpful.

Results

In lines 242 – 244, provide a brief description or example of what is meant by “sensory experiences” of an IC. Also, note that as a reviewer, I did not have access to the supplementary materials (to the best of my knowledge) and would be interested in seeing them.

Discussion

In lines 286 – 288, I am concerned that this sentence reads as a biased and western take on what creativity is and how it manifests across cultures. No doubt that China has produced some of the most celebrated and venerated narratives, music, visual art, dance, film, food and fashion in recorded human history.

In lines 289 – 292, please discuss with more clarity how recruitment methods may have inflated the number of past and current ICs reported, and how this limits study findings without replication. Please also be sure to mention the use of retrospective reporting of ICs as a limitation.

More minor points:

1. There are a few minor typos throughout the manuscript.

2. In Table 1, please report the ns in raw numbers as well as in percentages.

Reviewer #3: In the paper titled “A Cross-National Study Examining Imaginary Companions and Face Pareidolia in British and Chinese Adults”, the researchers are investigating the differences in creation of Imaginary companions in childhood and adulthood in Chinese and British adults, the differences in face pareidolia, and the link between creating an IC and pareidolia.

They collect data from 291 adults (185 Chinese) through online questionnaires. They did not find differences in the prevalence of adulthood ICs between Chinese and British adults, although British were more likely to report a childhood IC. In the face pareidolia task Chinese participants reported more false alarms than the British. Finally, those with a current IC were more likely report correct face pareidolia. These results are interpreted as support for the “role of top-down processing in both imaginary companion creation and hallucination.”

I find the research question interesting and the tasks employed as appropriate. However, I have questions about the conceptualization of the research question and the interpretation of the results.

The researchers discuss the research question in relation to hallucinations, however, I believe that the link between hallucinations and ICs (and pareidolia) needs to be explained/rationalized better in the introduction. The rationale behind the link between top-down processing and ICs are clear but I do find the link to hallucinations a little unclear.

It’s reported that “individual differences in the ability to detect face pareidolia have been found to correlate with the frequency of visual hallucination-like experiences in non clinical participants”. Here, I am confused as to what ‘visual hallucination-like experiences’ would be like. Can they also be considered signs of a person’s high creativity/imagination? How is the distinction made between creativity/imagination and ‘hallucination-like experiences’?

There were 12 questions regarding the ICs. I wonder how the answers to these questions were utilized. Did the researchers only use the questions asking whether whether the participant had a past or current IC, or did they also take into account participants’ answers to the other questions about sensory experiences, etc. when deciding whether what the participant is reporting a real IC or something else. This might especially be an issue when participants are reporting Personified Objects. Because, unless the answers to the other questions are taken into account, it is difficult to know whether the participants are reporting a real IC (PO) or whether they are simply thinking about their comfort and/or transition objects.

On the same questionnaire, participants were asked to rate themselves on creativity. Were the answers to these questions used in any analyses? It might be relevant and interesting, as the researchers discuss that “Collectivist cultures like China have traditionally been more likely to value harmony, conformity and consensus, which may be less creativity oriented” (p.13).

The age is only used as a categorical variable, which I think is okay. But I would like to see a reason for how they decided on these categories (e.g., 18-30 and 31-50). When the distribution in two samples (British vs Chinese) are examined in Table 1, we see that about 80% of Chinese sample was between 18 and 30. I would like the researchers to mention this difference and discuss (if relevant) in relation to the results.

On page 15, it’s stated that “Because ICs can be iICs or POs and both types can have any combination of auditory and visual components for their creators, there are questions around whether there are related differences in face or auditory verbal detection depending on whether one sees or hears their companion. It might be interesting for future research to investigate modality of IC experience and determine whether this relates to top-down processing in only visual tasks, but also auditory.” This question can partially be tested with this dataset, as the researchers are already collecting information about participants’ sensory experiences of their ICs (e.g., Do the participants who report ‘seeing’ their ICs perform differently on the pareidolia task?)

Minor points:

I wonder how the study was advertised and how the recruitment was done – e.g., was the call for participation specify ‘those with ICs’, etc.?

There are typos throughout the paper that need to be fixed – one major typo being in the running head.

On Page 11, it’s stated that “Nature of participants’ ICs allowed us to examine associations between performance on the pareidolia task and individual differences in, for example, the ‘sensory’ experiences of an IC. This analysis was exploratory and these associations are reported in Supplementary Materials.”. However, I could not access this file; the supplementary material file only included the SPSS dataset.

There are also some missing information in the references (e.g., for the paper by Majors and Baines (2017), the Journal Title is incorrect).

**Do you want your identity to be public for this peer review?** For information about this choice, including consent withdrawal, please see our Privacy Policy

Reviewer #1: No

Reviewer #2: No

Reviewer #3: No

---

## [Author Response · Author response to Decision Letter 1]

30 Sep 2025

Dear Editors,

Thank you for your review of the manuscript entitled, A Cross-National Study Examining Imaginary Companions and Face Pareidolia in British and Chinese Adults. We have amended the manuscript according to the feedback and have laid out our changes below. The main text is now 6291 words excluding the abstract and tables/figures. It is now 30 pages in length including the main unblinded text, references, tables and figures. It contains 4 tables 2 figures and appendices.

Sincerely,

Paige E. Davis, PhD, FHEA,

Lecturer in Psychology

University of Leeds

School of Psychology

LS2 9JT

+44 (0)7542 098888

p.e.davis@leeds.ac.uk

Reviewer #1

1. The conclusion of the abstract comes across as too broad and somewhat disconnected from the actual study scope. For example, the sentence: “Taken together, the results provide more information around the role of top-down processing in both imaginary companion creation and hallucination. This could be a way to learn about the developmental trajectory of hallucination from non-pathological to pathological.” seems overly speculative. Since the study is mainly of exploratory nature and focuses on ICs and face pareidolia, it does not directly investigate hallucinations, especially not pathological ones. For instance, Romagnano et al. (2022) examined face pareidolia in individuals with schizophrenia and found no significant differences in pareidolia perception between patients with or without hallucinations.

Thank you for this suggestion. We have now changed the conclusion to reflect what we found rather than going too broad with the implications. This is what we have:

Taken together, the results provide more information around imaginary companion creation in China and the UK as well as the role top-down processing may play in imaginary companion interactions.

2. Participants:

It would be helpful to clarify whether participants were screened in the questionnaire for psychiatric, developmental, or neurological conditions, given the significant impact such conditions can have on social cognition and related perceptual processes.

We have now added a sentence in the method stating:

Participants were not screened for any psychological, developmental or neurological conditions.

(lines 181-182)

As well as a sentence in the limitations using the wording suggested in this review. Thank you.

Another consideration is that participants were not screened for any psychiatric, developmental or neurological conditions and given the significant impact on social cognition and perceptual processes, having this information could improve our ability to evaluate the results. (lines 437-439)

3. Visualization:

The paper would benefit from improved visualization. Including examples of the stimuli used and/or graphical representations of key results would help readers better understand and interpret the findings.

We have now added a figure which shows the stimuli.

We have also added a figure which is a graphical representation of the pareidolia hits. We think this improves understandability of the findings.

4. Discussion:

The authors mention that they were surprised to find cultural differences between the UK and China in face pareidolia, referencing a study that found no significant differences between two European subcultures (reference 33). However, given the well-documented differences in face processing between Western and East Asian populations, differences in pareidolia perception might indeed be expected. This could be discussed in more depth, potentially referencing additional literature.

Additionally, Zhou and Meng (2021) analyzed face pareidolia using signal detection theory (SDT). Incorporating similar analyses, such as sensitivity (d') and decision criterion (c), could provide more insight into the nature of the cultural differences observed and should be considered either in this paper or in future work.

Thank you for bringing this issue up. After re-reading the paper about cultural differences we took the decision to refine the argument in the introduction and explain that the study (original manuscript reference 33) is in two European subcultures rather than Eastern and Western cultures. We also explained further about the large individual differences found in pareidolia. This set the argument up in the discussion to focus more on differences in face processing.

Because pareidolia tasks ask participants to view pictures and make assessments of objects resembling faces with no clear race or ethnicity, rather than real faces, these tasks have been used in cross cultural studies and Romagnano and colleagues have even attempted to rule out own-culture and gender biases when participants look at faces (31). However, Romagnano’s research examined two European cultures which were situated closely together. There have not been any studies to the authors knowledge on individual differences in pareidolia in Chinese and British Adults. Large individual differences in the experience of face pareidolia has been reported in many studies over domains like gender, developmental period, personality and neural mechanism (32,33). (line 145-153)

We also changed the discussion to explain that face processing differs with culture and made your point about this transferring to pareidolia. See below paragraph for the change:

When considering the pareidolia scores, there were cross-national differences in false alarms where Chinese participants made significantly more false alarms, but there were no differences in pareidolia hits. It is well documented that there are cultural differences in face processing (e.g.46). For example, in Europe, French-speaking Swiss participants and German participants had different patterns of face tuning (47). More diverse culture shifts comparing Canadian and Chinese participants show that even the way visual information about a face is received can be impacted by culture (48). It may follow that, with these cultural differences in face processing, there may also be differences in pareidolia perception. (lines 384-389)

We added Zhou and Meng (2020) and Wang and Yang (2025) to intro and Caldara (2017), Pavlova et al (2018) and Tardif et al (2017) into the discussion.

Thank you for the suggestion to include analyses involving signal detection parameters. We have calculated these values and have included analysis employing these parameters in the Supplementary Materials. Note that the pattern of effects when using d' and C are consistent with the main analyses. Group differences were not significant when we compared participants with versus without a childhood IC. Similarly, the group difference in d' was not significant when we compared participants with versus without a current IC. However, there was a significant group difference in C when we compared participants with versus without a current IC. Consistent with the greater number of ‘hits’ made by participants with a current IC, they also had a lower mean C-score (i.e., they had a more liberal bias, towards responding that a face was present in the images).

Reviewer #2-We appreciate that no AI tools were used and thank you for taking the time to do this yourself.

1. Overall Changes:

I am struggling to understand the use of face pareidolia as a proxy for hallucinations, particularly given the task that was used to measure face pareidolia in this study. Hallucinations, to the best of my understanding, are hearing, seeing or experiencing things for which there is no basis of reality, or for which there is no objective evidence of existence. With hallucinations, what is “real” or “true” appears to be a key consideration in measuring what is and is not a hallucination (at least in western cultures). However, seeing faces in visual stimuli such as abstract images, clouds, objects, etc. is a subjective experience in which a basis of reality may not matter as much, and to the best of my knowledge, more closely associated with individual differences in anthropomorphism in the literature. With abstract imagery, the presence of a face in visual stimuli could simply be “in the eyes of the beholder.” In other words, seeing a face in abstract visual imagery is neither “true” nor “false”; it just is.

And that brings me to my concern about the task designed to measure face pareidolia, where “correct hits” are described as suggestive of a greater propensity for hallucinations, and “false positives” are discussed simply as errors. If the task was designed as a measure or proxy for the experience of hallucinations, then the “false positives” are of greater interest than the “correct hits.” And in this study, there were no differences in false positives based on IC status (past or current, absent or present), which would change the authors’ interpretation of the findings. However, if the task is not intended to serve a proxy for hallucinations but rather a direct measure of face pareidolia alone, then it is unclear why the measure needed to have “correct” or “incorrect” responses. In other words, even with the same task, why not simply measure for each participant the number of images in which a face was detected?

Apologies that this was not clearer in the original manuscript. The pareidolia task is not meant to measure/be a proxy for experiencing hallucinations. Instead, the task aims to measure ‘top-down processing’ (i.e., the extent to which perception is shaped by expectations) – a process that we think plays a role in the development of both ICs and hallucinations. We have tried to make this clearer by editing a sentence in the Method section:

The face pareidolia task designed by Smailes and colleagues was used to measure top-down processing presented to the participants through the Gorilla platform. (lines 228-230)

The reviewer is right that, to some extent, face pareidolia are ‘in the eye of the beholder’. However, to make this task less ambiguous we developed it in the following way, so that we think it is appropriate to consider some responses to be ‘correct’ and some to be ‘incorrect’: The presence or absence of a face pareidolia in the images presented to participants was established in a pilot study. In this pilot, each image from the final task was presented to a group of seven participants (who did not take part in the full study). Images were presented using Microsoft PowerPoint, in a fixed, random order. Participants were given unlimited time to detect whether or not a face pareidolia was present or absent in each image. All participants detected the face pareidolia in the 24 images that we have classed as “pareidolia present”. None of the participants detected a face pareidolia in the12 images that we have classed as “pareidolia absent”. We have added this information about how the task was developed into the Supplementary Materials.

RE: The point about ‘pooling’ participants ‘face-present’ responses (i.e., summing ‘hits’ and ‘false alarms’). To some extent, the additional analyses we have performed (and included in the Supplementary Materials) using C-score as an outcome variable achieves this (as C-score is calculated using both hits and false alarms). We have, however, left this analysis as supplementary and have focused on the ‘raw’ hits and false alarms because we believe that hits and false alarms measure different processes. That is, we think that individual differences in top-down processing are measured by the number of hits a participant makes. However, we do not think that individual differences in top-down processing are measured by the number of false alarms a participant makes. Consistent with this claim, the correlation between the number of hits and number of false alarms a participant makes was medium-sized in this study (rho = .326) and was close-to-zero in a previous study that one of us (DS) conducted (rho = .07; see these supplementary materials). If both number of hits and number of false alarms both measured top-down processing, we would expect these correlations to be stronger.

2. Suggestions: In the literature review, make a stronger case as to how seeing faces in visual stimuli has links to hallucinations and to ICs, from a theoretical perceptive and in terms of measurement. Reconsider the description of the measure of face pareidolia (i.e., if it is indeed being used as a proxy for measuring hallucinations or if is not meant to imply anything beyond face pareidolia). If it is being used as a proxy for measuring hallucinations, please explain why “false positives” were not the primary variable of interest.

We have edited the Introduction so that it is hopefully clearer that our logic is as follows: 1 = ICs and hallucinations/hallucinatory experiences (H/HEs) are similar in some ways; 2 = They may be similar in that the same processes may play a causal role in the development of both ICs and H/HEs; 3 = Past research has shown that hearing words in ambiguous speech sounds (which probably reflects elevated top-down processing in the auditory domain) is associated with both ICs and H/HEs; 4 = Visual H/HEs are associated with a tendency to detect more face pareidolia (which probably reflects elevated top-down processing in the visual domain); 5 = Given the ICs often have a visual component, it seems plausible that having an IC will be associated with elevated top-down processing in the visual domain; 6 = We, therefore, expected that participants with an IC would detect more face pareidolia in the task than would participants without an IC.

RE: the comment about why false alarms were not our primary outcome measure – please see our response above.

3. Literature Review

Again, in the literature review, I am struggling to understand how top-down processing, ICs, hallucinations, and face pareidolia are all theoretically connected. Consider adding a schematic image to clarify for the reader.

Thank you for this idea, we have edited the Introduction so that it is hopefully clearer how we think these factors are related.

4. The literature review also assumes prior knowledge of other ideas/constructs, such as theory of mind and top-down processing. Provide brief definitions so that readers who are not as familiar with this literature can understand the points that are being made.

Thank you for this comment. The ideas of theory of mind and top-down processing now have been explained in the introduction.

This elevated performance in tasks around theory of mind, or the understanding that others can take different perspectives, as well as having different thoughts feelings or ideas, has been replicated in China. (lines 105-107)

that a person’s tendency to detect speech sounds in ambiguous auditory stimuli reflect individual differences in auditory top-down processing (processing using prior knowledge or expectations to interpret ambiguous stimuli) (lines 123-125)

5. In lines 98 – 102, it would be helpful to understand the interplay between cross-cultural differences in face pareidolia and ICs. Perhaps speculate as to what this interaction might be or look like in this section. In addition, more information about how face pareidolia tasks have been used to assess cultural and gender biases would be helpful.

Please see 2nd comment of your review. We are hoping that this is now clear.

We have now added in the task used to look at pareidolia and culture in the Romagnano and colleagues paper.

Because pareidolia tasks ask participants to view pictures and make assessments of objects resembling faces with no clear race or ethnicity, rather than real faces, these tasks have been used in cross cultural studies and Romagnano and colleagues have even attempted to rule out own-culture and gender biases when participants look at faces (31). However, Romagnano’s research examined two European cultures which were situated closely together. There have not been any studies to the authors knowledge on individual differences in pareidolia in Chinese and British Adults. Large individual differences in the experience of face pareidolia has been reported in many studies over domains like gender, developmental period, personality and neural mechanism (32,33). (line 145-152)

6. Also, in the literatur

---

## [Decision Letter · Decision Letter 1]

28 Oct 2025

Dear Dr. Davis,

Reviewer 3 has highlighted some outstanding points that need to be addressed before reconsideration for publication.

We look forward to receiving your revised manuscript.

Kind regards,

Clare Eddy

Academic Editor

PLOS ONE

Journal Requirements:

Reviewers' comments:

Reviewer's Responses to Questions

**Comments to the Author**

Reviewer #1: All comments have been addressed

Reviewer #2: All comments have been addressed

Reviewer #3: (No Response)

2. Is the manuscript technically sound, and do the data support the conclusions?

Reviewer #1: Yes

Reviewer #2: Yes

Reviewer #3: Partly

3. Has the statistical analysis been performed appropriately and rigorously?

Reviewer #1: N/A

Reviewer #2: Yes

Reviewer #3: Yes

4. Have the authors made all data underlying the findings in their manuscript fully available?

Reviewer #1: Yes

Reviewer #2: Yes

Reviewer #3: No

5. Is the manuscript presented in an intelligible fashion and written in standard English?

Reviewer #1: Yes

Reviewer #2: Yes

Reviewer #3: Yes

Reviewer #1: (No Response)

Reviewer #2: I appreciate the authors' through care and attention to my comments and recommended revisions. I have no further recommendations at this time.

Reviewer #3: I thank the authors for taking into account many of the comments that were brought up in the earlier version of the manuscript.

However, I still have some remaining concerns regarding both theoretical and methodological aspects of this study.

*As I raised a concern in my previous review, I still think that the expected link between hallucinations and ICs (and pareidolia) needs to be rationalized better in the introduction. In the current version of the manuscript, on page 4 the authors state that “…These links between experiencing ICs and individual differences in theory of mind and creativity, as well as similarities in their phenomenology, have resulted in some researchers (e.g., [20, 23]) suggesting that there are important parallels between IC and hallucinations.” I believe the authors could make their reasoning more explicit.

*Throughout the text, the authors refer to sensory experiences of ICs and how they might be related to hallucinatory experiences. However, as POs are also considered imaginary companions, it would be important to distinguish between invisible ICs and POs, and make separate arguments for their findings when necessary (e.g., visual hallucinations might not be relevant to POs).

*This is also related to my comment in previous version regarding the analysis in terms of sensory experiences. Seeing a PO is not hallucinatory and does not require top-down processing, and given that many adults in the current study report current POs rather than current invisible ICs, I think this should be acknowledged.

*On Page 6, it reads “Romagnano and colleagues have even attempted to rule out own-culture and gender biases when participants look at faces”. Here I find it difficult to relate this Romagnano study to the current research question. I believe further information about the Romagnano (2002) study would be helpful.

*I would like to see a more elaborate discussion of the finding that Chinese participants make significantly more false alarms on the task. In discussion section, on page 16, the authors refer to this result and talk about cultural differences in face processing (“It may follow that, with these cultural differences in face processing, there may also translate to differences in pareidolia perception” lines 388-89). I am having a hard time understanding how this result can be explained away with these. Because 1) we do not see cultural differences in hits, 2) the authors stateon page 17, line 394, the authors write “The greater number of false alarms made by Chinese participants may reflect cross-national differences in response biases (49), rather than reflecting differences in cognitive processes, or may simply be statistical ‘noise’.”

*On page 15, the researchers state that “In our research, British adults remembered significantly more ICs in childhood than Chinese adults. One possible explanation for this difference could be that British culture has been traditionally considered to be individualistic, valuing independence, novelty, autonomy and freedom, all values which may lead to creation of an imagined social relationship”. This could very well be true, however, researchers also report that there is no difference in adult creation of ICs and it seems like that their questionnaire taps at remembering of past Ics rather than necessarily of ‘existence of a childhood IC’. Then, I believe that one should be cautious when interpreting results regarding childhood vs. adult IC. When participants report having a childhood IC, we are actually tapping their memory of one (e.g., they might have forgotten that they had one – especially if it’s not talked much in the family), and it might not mean the same thing as when children report having an IC.

Minor point:

• Gender can be added to the Table 1, to demonstrate basic demographics.

• The SPSS file was available as supplementary material in the previous version, but not in the current version.

**Do you want your identity to be public for this peer review?** For information about this choice, including consent withdrawal, please see our Privacy Policy

Reviewer #1: No

Reviewer #2: No

Reviewer #3: No

---

## [Author Response · Author response to Decision Letter 2]

26 Nov 2025

Reviewer #1

1. No response

Reviewer #2

1. I appreciate the authors' through care and attention to my comments and recommended revisions. I have no further recommendations at this time.

We appreciate the time and effort you've put into peer review, thank you.

Reviewer #3

1. *As I raised a concern in my previous review, I still think that the expected link between hallucinations and ICs (and pareidolia) needs to be rationalized better in the introduction. In the current version of the manuscript, on page 4 the authors state that “…These links between experiencing ICs and individual differences in theory of mind and creativity, as well as similarities in their phenomenology, have resulted in some researchers (e.g., [20, 23]) suggesting that there are important parallels between IC and hallucinations.” I believe the authors could make their reasoning more explicit.

Apologies that this remained unclear. We hope that we have ‘fleshed out’ our reasoning more clearly in the revised manuscript (lines 112-119), where we explain how ICs and hallucinations can be similar in phenomenological terms and may be similar in terms of the factors that play a causal role in their development.

2. *Throughout the text, the authors refer to sensory experiences of ICs and how they might be related to hallucinatory experiences. However, as POs are also considered imaginary companions, it would be important to distinguish between invisible ICs and POs, and make separate arguments for their findings when necessary (e.g., visual hallucinations might not be relevant to POs).

We now have written a paragraph in the discussion about this and what future research could investigate (lines 445-452):

Another consideration around iICs and POs is that the experience children are having with them might later result in differences in pareidolia. We theorise that children are using the same type of imagination processing with iICs as with POs. As Taylor and Aguiar argue, children might very well be utilizing their own imagination to embellish the appearance of their PO so that it may look quite different in their mind's eye than in reality (8). Asking participants for details around what their PO looks like to them might be a question to ask adult participants in future research. If it were true that iICs and POs were different imagination processing experiences, the results of many IC studies including this one would need to be reconsidered.

3. *This is also related to my comment in previous version regarding the analysis in terms of sensory experiences. Seeing a PO is not hallucinatory and does not require top-down processing, and given that many adults in the current study report current POs rather than current invisible ICs, I think this should be acknowledged. We have now acknowledged this in the discussion (lines 442-444).

Parsing apart iICs from POs is especially important with the number of POs in the study sample. Because POs can be seen and may not relate as much to top-down processing, there could be another reason for the results.

4. *On Page 6, it reads “Romagnano and colleagues have even attempted to rule out own-culture and gender biases when participants look at faces”. Here I find it difficult to relate this Romagnano study to the current research question. I believe further information about the Romagnano (2002) study would be helpful.

Apologies that this was not clearer in the previous submission. This was likely due to a typo (‘faces’ versus ‘face pareidolia’). In lines 152-163, we have explained that previous findings re: cultural differences in face processing may not be found when studying face pareidolia because the ‘faces’ being processed by participants are much simpler than ‘real faces’. We note that Romagnano et al. (2022) examined this question and did not find cross-cultural differences in face pareidolia detection, but that their cross-cultural comparison involved groups from northern versus southern Europe. And so there is value in examining other cross-national (i.e., UK versus China) differences in face pareidolia.

5. *I would like to see a more elaborate discussion of the finding that Chinese participants make significantly more false alarms on the task. In discussion section, on page 16, the authors refer to this result and talk about cultural differences in face processing (“It may follow that, with these cultural differences in face processing, there may also translate to differences in pareidolia perception” lines 388-89). I am having a hard time understanding how this result can be explained away with these. Because 1) we do not see cultural differences in hits, 2) the authors stateon page 17, line 394, the authors write “The greater number of false alarms made by Chinese participants may reflect cross-national differences in response biases (49), rather than reflecting differences in cognitive processes, or may simply be statistical ‘noise’.”

Apologies that our previous argument(s) wasn’t clearer. In lines 405-429, we have tried to make our argument more effectively. We note that (in agreement with the reviewer), it seems unlikely that cross-national/cultural difference in face-processing would drive this result. In addition we think it is unlikely that a pareidolia-specific process (i.e., top-down processing) would drive this result.

And so, instead, we propose that it is possible that cross-cultural differences in more generic response biases or in metacognitive confidence may drive this result. However, we note that we know nothing about how generic response biases and/or metacognitive confidence shape performance on the face pareidolia task, so it is hard to be confident that this is a good explanation.

We also expand on the suggestion that the finding may simply be noise/a Type 1 error. We note that the small p-value associated with our finding might be interpreted as this being a weak argument, but that this interpretation is wrong, as a single p-value can’t really be used in that way.

6. *On page 15, the researchers state that “In our research, British adults remembered significantly more ICs in childhood than Chinese adults. One possible explanation for this difference could be that British culture has been traditionally considered to be individualistic, valuing independence, novelty, autonomy and freedom, all values which may lead to creation of an imagined social relationship”. This could very well be true, however, researchers also report that there is no difference in adult creation of ICs and it seems like that their questionnaire taps at remembering of past Ics rather than necessarily of ‘existence of a childhood IC’. Then, I believe that one should be cautious when interpreting results regarding childhood vs. adult IC. When participants report having a childhood IC, we are actually tapping their memory of one (e.g., they might have forgotten that they had one – especially if it’s not talked much in the family), and it might not mean the same thing as when children report having an IC. We have added this caution and some research that backs up the claim (lines 368-371):

Another possibility is that the questionnaire focused on remembering an IC rather than actually having one. Retrospective report of an IC has been found to be inaccurate in adolescents and misremembering may be a reason to interpret the results around childhood ICs with caution (45).

7&8.

• Gender can be added to the Table 1, to demonstrate basic demographics.

• The SPSS file was available as supplementary material in the previous version, but not in the current version. - We have added gender to table 1 (lines 638-642)

- We were told we could not attach the new SPSS file because it couldn't be opened for the last revision. The corresponding author has now emailed to make certain we can put the updated SPSS file into this revision. Thank you for noting this as it made it possible to contact the editor and query the issue.

---

## [Decision Letter · Decision Letter 2]

17 Dec 2025

Dear Dr. Davis,

Thank you for submitting your manuscript to PLOS ONE. After careful consideration, we feel that it has merit but does not fully meet PLOS ONE’s publication criteria as it currently stands. Therefore, we invite you to submit a revised version of the manuscript that addresses the points raised during the review process.

In addition to the points raised by the reviewer #3 elsewhere within this email, please see further required corrections immediately below.

The abstract must be understandable on its own. Please re-write L37-38 as the implication is unclear. It is also unclear what the meaning of “correct face pareidolia” is.

L113 – The word ‘typically’ may be changed to ‘sometimes’

L153 – Images used in all pareidolia tasks do not always resemble faces. Sometimes they are simply noise, in order to elicit false alarms. Please ensure statements such as this are accurate.

L368 – This needs to be rewritten. Is the word ‘because’ missing?

L400 – This paragraph begins with a clause related to the previous sentence. Please structure appropriately.

L397 – Explain what is meant by “received”

L421 – This sentence should be restructured.

After implementing these corrections and those noted by reviewer #3, please read through your entire manuscript carefully, as if you are someone who is unfamiliar with the study and topic, and ensure that all language is precise, and that what you mean is clear to readers.

I will then make a final decision as to suitability for publication.

We look forward to receiving your revised manuscript.

Kind regards,

Clare Eddy

Academic Editor

PLOS One

Journal Requirements:

Reviewers' comments:

Reviewer's Responses to Questions

**Comments to the Author**

Reviewer #3: All comments have been addressed

2. Is the manuscript technically sound, and do the data support the conclusions?

Reviewer #3: Yes

3. Has the statistical analysis been performed appropriately and rigorously?

Reviewer #3: Yes

4. Have the authors made all data underlying the findings in their manuscript fully available?

Reviewer #3: Yes

5. Is the manuscript presented in an intelligible fashion and written in standard English?

Reviewer #3: Yes

Reviewer #3: I thank the authors for taking into account my comments and recommended revisions.

A few minor issues (mainly typos) that I thought the authors might like to address before publication:

1) Typo on Line 169- I think it should read adults' (with apostrophe)

2) Line 187 - may say 'age and gender data for ...'

3) Typo on Line 303 -"in contrast..." ("I" should be capitalized)

4) statistical letters should be italicized in when reporting results (e.g., r, p)

5) there is a large overlap between the info provided in Table 4 and Figure 2. They could be integrated or one can be omitted.

**Do you want your identity to be public for this peer review?** For information about this choice, including consent withdrawal, please see our Privacy Policy

Reviewer #3: No

---

## [Author Response · Author response to Decision Letter 3]

18 Dec 2025

17th December, 2025

Dear Editors,

Thank you for your third review of the manuscript entitled, A Cross-National Study Examining Imaginary Companions and Face Pareidolia in British and Chinese Adults. We have amended the manuscript according to the feedback you and the reviewer #3 have suggested. We have laid out our changes below. The main text is now 6871 words excluding the abstract and tables/figures. It is now 32 pages in length including the main unblinded text, references, tables and figures. It contains 4 tables 1 figure and appendices.

Sincerely,

Paige E. Davis, PhD, FHEA,

Lecturer in Psychology

University of Leeds

School of Psychology

LS2 9JT

+44 (0)7542 098888

p.e.davis@leeds.ac.uk

Editor

1. The abstract must be understandable on its own. Please re-write L37-38 as the implication is unclear. It is also unclear what the meaning of “correct face pareidolia” is.

We have re written the sentence to make more sense taking out the word 'correct' as we agree, it could be unclear not having read the study. L37-38

L113 – The word ‘typically’ may be changed to ‘sometimes’

This has now been changed. Thank you. L113

L153 – Images used in all pareidolia tasks do not always resemble faces. Sometimes they are simply noise, in order to elicit false alarms. Please ensure statements such as this are accurate.

These statements are accurate about face pareidolia tasks including the one we used.

L368 – This needs to be rewritten. Is the word ‘because’ missing?

We have now changed the sentence to say …which may result in less individual ICs (45). However, with more movement around the globe… L367

L400 – This paragraph begins with a clause related to the previous sentence. Please structure appropriately.

We took out the word 'however' here and started with We think… to structure the sentence more appropriately. L400

L397 – Explain what is meant by “received”

We have changed this word to perceived. L398

L421 – This sentence should be restructured.

This sentence has been broken up into two sentences and made more clear. L420-422.

Reviewer #3

1. Typo on Line 169- I think it should read adults' (with apostrophe)

This has now been amended. Thank you for that catch. L169

2) Line 187 - may say 'age and gender data for ...' We have added this now. Thank you. Ll87

3) Typo on Line 303 -"in contrast..." ("I" should be capitalized)

The I is now capitalised L303

4) statistical letters should be italicized in when reporting results (e.g., r, p)

Apologies for this. The statistical letters are now italicised throughout the results section.

5) there is a large overlap between the info provided in Table 4 and Figure 2. They could be integrated or one can be omitted. We have now cut Figure 2 as it has less information.

---

## [Editor Report · Decision Letter 3]

21 Dec 2025

A Cross-National Study Examining Imaginary Companions and Face Pareidolia in British and Chinese Adults

PONE-D-25-25109R3

Dear Dr. Davis,

We’re pleased to inform you that your manuscript has been judged scientifically suitable for publication and will be formally accepted for publication once it meets all outstanding technical requirements.

Kind regards,

Clare Eddy

Academic Editor

PLOS One

**Additional Editor Comments** (optional):

At R3 stage the authors were requested to thoroughly review the manuscript in case of remaining errors/typos. There are still errors within the manuscript e.g. typing error within the running head; use of the word 'less' rather than 'fewer'; poor ordering of clauses in the first sentence of the abstract; the paragraph with an inappropriate break (L399) because after the break it continues to discuss the same point, etc. While the manuscript is now accepted for publication, I am recommending again to the authors that they check and amend their manuscript carefully (now at proofing stage), as I assume there is an appreciation of the benefits of doing so.
---

## [Editor Report · Acceptance letter]

PONE-D-25-25109R3

PLOS One

Dear Dr. Davis,

I'm pleased to inform you that your manuscript has been deemed suitable for publication in PLOS One. Congratulations! Your manuscript is now being handed over to our production team.

Kind regards,

on behalf of

Dr. Clare Eddy

Academic Editor

PLOS One